# Use of Vibrational Optical Coherence Tomography to Analyze the Mechanical Properties of Composite Materials

**DOI:** 10.3390/s21062001

**Published:** 2021-03-12

**Authors:** Frederick H. Silver, Nikita Kelkar, Tanmay Deshmukh

**Affiliations:** 1Department of Pathology and Laboratory Medicine, Robert Wood Johnson Medical School, Rutgers, The State University of New Jersey, Piscataway, NJ 08854, USA; 2OptoVibronex, LLC., Allentown, PA 18104, USA; nuk1@scarletmail.rutgers.edu (N.K.); tmd24895@gmail.com (T.D.)

**Keywords:** energy storage, energy dissipation, modulus, resonant frequency, elastic behavior, composite materials, flexibility, collagen, silicone rubber, Viton rubber

## Abstract

Energy storage and dissipation by composite materials are important design parameters for sensors and other devices. While polymeric materials can reversibly store energy by decreased chain randomness (entropic loss) they fail to be able to dissipate energy effectively and ultimately fail due to fatigue and molecular chain breakage. In contrast, composite tissues, such as muscle and tendon complexes, store and dissipate energy through entropic changes in collagen (energy storage) and viscous losses (energy dissipation) by muscle fibers or through fluid flow of the interfibrillar matrix. In this paper we review the molecular basis for energy storage and dissipation by natural composite materials in an effort to aid in the development of improved substrates for sensors, implants and other commercial devices. In addition, we introduce vibrational optical coherence tomography, a new technique that can be used to follow energy storage and dissipation by composite materials without physically touching them.

## 1. Introduction

Composite materials make up the majority of the structural components found in the human body as well as in materials of construction for sensors and other industrial applications. The ability to measure the mechanical properties of composite materials is needed to understand the differences between normal and diseased tissues as well as to define engineering design criteria for construction materials. Unfortunately, defining tensile, compressive or bulk moduli and ultimate tensile strengths non-invasively and non-destructively for materials without modifying their properties is difficult. While indentation tests offer some measure of the material’s stiffness, they are unable to measure the moduli of the components of composite materials as well as the energy storage and dissipative abilities.

Energy storage and dissipation mechanisms are important parameters for both tissues and synthetic materials. Examples include the storage, transmission and dissipation of energy in the musculoskeletal system in mammals [1,2,3,4,5,6,7] and the ability of automotive gaskets and bushings to undergo millions of loading and unloading cycles limiting fatigue failure.

The molecular mechanisms of energy storage in the musculoskeletal system have been advanced based on the results of studies on race horses and other animals [3,4,5,6], molecular modeling and mechanical testing of collagen fibers [2]. The results of these studies suggest that muscle contraction provides a force that does work on the attached tendons [3,4,5,6] leading to elastic strain energy storage in the collagenous structural components [1,2,7]. Conformational and free energy changes in collagen triple helices at the molecular level act to store energy but also provide a means for energy distribution to other parts of the musculoskeletal system [1,2]. Energy dissipation that occurs in skin at low strain rates [7] involves reversible rearrangement of fluid and polymeric molecules that occupy the spaces between collagen fibers of the dermal structural components [7,8,9]. To facilitate design of new energy storage and dissipative devices requires further study of these properties using methods that do not modify the substrates during testing.

A number of tests have been used to characterize tissues and materials including constant rate-of-strain experiments where the modulus (E) is obtained from the slope of the stress-strain curve and does not require assuming a value of Poisson’s ratio [10]. The determination of the stiffness or modulus requires measurements made at several different levels of the strain, and the slope (E) depends on the rate of deformation. The limitations to this test are as follows: (a) the sample is destroyed during testing; (b) the results need to be corrected for strain-rate dependence; (c) the modulus can only be evaluated from the slope of the stress-strain curve, which requires measurements at several increasing levels of the strain; and (d) the value of E is difficult to determine from the slope of the stress-strain curve when the slope is rapidly changing as is the case for tissues [11]. 

Because of the limitations of the constant rate-of-strain tests and the invasiveness of the test, a variety of new approaches have been developed in the last two decades to attempt to characterize the mechanical properties of tissues and materials. These include magnetic resonance elastography, ocular response analysis, optical coherence tomography, ultrasound elastography, surface wave optical coherence elastography and vibrational optical coherence tomography [11,12,13,14,15,16,17,18,19,20].

### Vibrational Optical Coherence Tomography

Recently, we reported a new method to measure elastic and viscous properties of materials non-invasively and non-destructively using a technique termed vibrational optical coherence tomography (VOCT) [8,9,20,21,22,23,24]. This technique uses audible sound to vibrate a material transverse to the surface and infrared light that is reflected off the surface to measure the resultant material displacement as a function of frequency [20,21,22,23,24]. The modulus is then calculated from the frequencies at which the maximum displacement is observed using the material thickness. This method can also be used to measure the viscous loss as a function of frequency [8]. Energy storage and viscous loss of the major components of composite materials can be evaluated based on results of VOCT. 

In this paper we present the results of VOCT studies on natural polymeric tissue composites and synthetic polymers to define the parameters that promote composite energy storage and dissipative properties. 

## 2. Methods

### 2.1. Measurement of Resonant Frequency and Elastic Modulus

VOCT is a non-invasive and non-destructive method that uses audible sound from a speaker to create a material displacement as described previously [20,21,22,23,24]. The displacement is detected using reflected infrared light that is analyzed by optical coherence tomography (OCT). The device consists of a modified OCT scanner operating at a wavelength of 840 nm with a scan rate of 13,000 lines per second in the A mode. A speaker provides a sinusoidal sound wave at increasing frequencies from a distance of about 1 inch above the tissue. The displacement peaks as a function of frequency are measured at a single point with the OCT operating in the B mode. The resolution of the resonant frequency measurements is 0.23 mm, and data are analyzed in the amplitude mode. The result is a mechanical spectrum of displacement as a function of frequency that defines the characteristics of each structural component of tissues or synthetic materials. The resonant frequency of a component is defined as the frequency at which the maximum displacement is observed. The measured resonant frequencies of composite materials are converted into modulus values using a calibration equation similar to Equation (1) for soft tissues developed based on in vitro uniaxial mechanical tensile testing and VOCT measurements made on the same material [20,21,22,23,24]. Tensile and vibrational modulus measurements made in vitro are used to develop Equation (1) for soft tissues and polymers. Equation (1) is an empirical equation that relates material resonant frequency and thickness to the modulus. Since most soft tissues have a density very close to 1.0, Equation (1) is valid for the majority of tissues found in the body; where the thickness d is determined from OCT images, $fn^{2}$ is the square of the resonant frequency, and *E* is the elastic modulus in MPa. The elastic modulus is a measure of the energy storage ability of a material.

Soft Tissues:(1)$$E*d=0.0651*\left({\mathrm{fn}}^{2}\right)+233.16$$

The resonant frequency or frequencies of each component of the sample are determined by measuring the displacement of the tissue resulting from sinusoidal driving frequencies ranging from 30 Hz to 20,000 Hz, in steps of 10 to 100 Hz. The maximum displacement occurs at the resonant frequency, fn^2^ [8,9,20,21,22,23,24]. The resonant frequencies and moduli of human tissues measured using VOCT are listed in Table 1 [9]. Table 1 is used to identify the tissue components resonating in a composite tissue.

### 2.2. Measurement of Loss Modulus as a Percent of the Elastic Modulus

Viscous loss measurements, which are related to the energy dissipation of a material, are reported as a percent of the elastic modulus. Samples are subjected to three pulses of audible sound waves at frequencies between 50 and 500 Hz in steps of 20 Hz. The viscous component of the viscoelastic behavior in MPa is obtained from the driving frequency peak by dividing the change in frequency at the half height of the peak (i.e., 3 db down from maximum peak in power spectrum) by the driving frequency after the third pulse has ended. This method is known as the half-height bandwidth method discussed by Paul Macioce (www.roush.com/wp-content/uploads/2017/08/Insight.pdf (Access on 5 August 2017)) [8].

## 3. Results

Figure 1 shows a plot of weighted displacement versus frequency for decellularized human dermis composed primarily of type I collagen [8]. The resonant frequency is defined as the frequency at which the maximum displacement occurs. The resonant frequency of decellularized dermis ranges from about 100 to 150 Hz, depending on the sample thickness, and the modulus at 5% strain is about 2 MPa [8,9].

Figure 2 shows a plot similar to Figure 1 for human skin above the biceps muscle. Note there are major peaks at 50, 120, 140, 380 and 440 Hz which represent cells (50), collagen fibers (120), blood vessels (140), muscle (380) and tendon (440). These peaks and the moduli are listed in Table 1. Note the spectrum of major peaks define the composition and the moduli of the major components. These measurements were made on the skin over muscle, blood vessels and attached tendons so vibrational peaks from all of these components are seen in Figure 2. Note that muscular tissue is rich in blood vessels, explaining the peak at 140 Hz.

Figure 3 shows a plot of weighted displacement versus frequency (Hz) for a crosslinked silicone rubber sample. Note the sharp resonant frequency peak at 120 Hz suggesting that the polymer chains are fairly monodisperse in length. The modulus of the silicone rubber sample was calculated to be about 1.7 MPa.

In contrast, the weighted displacement versus frequency plot shown in Figure 4 for new and fatigued Viton rubber gaskets shows much broader peaks compared to silicone rubber. The peak for new Viton rubber shifts to lower frequencies after cycling. The broadness of the peaks suggests that the polymer chains before and after cycling are polydisperse and that the average chain length decreases after cycling.

Figure 5 shows a plot of loss modulus as a percentage of the elastic modulus as a function of the frequency for decellularized dermis. The loss modulus was found to vary from about 3% to 12% and was almost independent of frequency at frequencies above 200 Hz. The loss modulus minimizes at about 100 Hz, the resonant frequency of collagen fibers, as previously reported [8,9]. In contrast, at low frequencies the viscous loss for skin is much higher than that of decellularized human dermis (Figure 6). This is attributed to fluid and proteoglycan rearrangement in the interfibrillar matrix under applied loads, which results in energy dissipation [9].

Figure 7 and Figure 8 illustrate that the loss modulus for silicone rubber and a new Viton rubber gasket were much lower than that of skin but are similar to that of decellularized dermis. These results suggest that polymeric materials are efficient in energy storage, but they do not dissipate energy very well. Figure 8 also shows that the percent loss modulus for a fatigued Viton rubber gasket increased above the value seen for the new gasket. As the elastic modulus decreased with increased cycling, the loss modulus appeared to increase, suggesting that increased energy dissipation may be a result of material fatigue and leads to premature mechanical failure.

## 4. Discussion

The design of composite materials that store energy elastically and dissipate it internally can be improved by analyzing the results of natural designs, especially in the human musculoskeletal tissues. It is well known that crosslinked random chain rubbery polymeric materials store energy elastically by uncoiling and increasing the end-to-end distance of the molecular chains [25]. While this is an efficient means to store energy at the molecular level, it is not as effective a mechanism by which the stored energy can be converted into work since the efficiency of polymeric motors is low. However, musculoskeletal and dermal tissues store and dissipate energy efficiently during locomotion.

### 4.1. Mechanical Properties of Model Musculoskeletal and Dermal Composite Materials 

Tendon and skin are examples of composite tissues that store energy and dissipate it efficiently [26,27]. Experimental results suggest that the values of the modulus of these tissues measured at both the fibrillar and the tissue levels vary greatly and change rapidly just prior to the onset of locomotion and loading [25,26,27,28]. The ability of materials to store energy is related to the modulus and the ductility.

McBride et al. [28] reported that the ultimate tensile strength (UTS) of developing chick extensor tendons increases from about 2 MPa at 14 days of development to 60 MPa 2 days after birth. This rapid increase in UTS is not associated with changes in fibril diameter but is associated with increases in collagen fibril lengths [28], which can be related to the viscoelastic properties of tendons [27]. While collagen fibrils clearly provide a means to store energy during loading, the properties of these elements are difficult to measure in vivo, and the results reported for values measured in vitro vary greatly.

The modulus of collagen fibrils in tendon varies from about 40 MPa for measurements made at low strains to 1.23 to 1.9 GPa for individual collagen fibrils [29,30,31,32,33,34,35,36,37]. In contrast, the reported Young’s modulus value for patellar tendons from young individuals is 660 MPa compared to 504 MPa for donors more than 64 years of age [31], while the reported value for the modulus of the patellar tendon in vitro in the highest force region is 1.3 GPa [32], which is much higher than the value of 34 MPa reported for in vivo measurements [9]. Clearly, these variations in reported moduli values for tendons establish the need for the development of new methods to non-invasively measure tendon modulus in vivo since in vitro test results vary greatly.

In a similar fashion, the mechanical properties of skin vary greatly as reported in the literature [38,39,40,41,42]. Daly [38] reported that at low strains the tensile modulus of skin is 5 kPa. Other study results suggest that the Young’s modulus of skin varies between 0.42 MPa and 0.85 MPa for torsion tests, 4.6 MPa and 20 MPa for tensile tests and between 0.05 MPa and 0.15 MPa for suction tests. Indentation test average values for Young’s modulus of skin are between 4.5 kPa and 8 kPa.

Clearly, evaluating the energy storage and dissipation properties of tendon and skin requires measurement of moduli of the individual components of tissues. The use of VOCT makes analysis of tissue energy storage and dissipation capabilities possible considering the non-linearity of the stress-strain curves for these tissues. However, it is first necessary to understand the structural basis for the stress-strain behavior of musculoskeletal and dermal tissues.

### 4.2. Stress–Strain Curves for Musculoskeletal and Dermal Tissues

The variation of reported stress-strain properties for tendon, skin and other non-mineralized tissues is due to the non-linearity of the in vitro stress-strain curves as well as differences in the degree of orientation of the collagen fibers in each tissue [41]. Figure 9 is a diagrammatic representation of the stress-strain curve of collagenous tissues in vitro. They exhibit a low modulus region where the modulus is relatively small and a high modulus linear region where it increases rapidly [41]. The upward curvature of the stress-strain curve makes in vitro analysis of the tensile properties of these tissues difficult since the modulus changes with the value of the strain. This explains some of the large variation in modulus values reported for tendon and skin; another reason for the range of values reported for the modulus is the dependence of the modulus on the strain rate [41]. In comparison to synthetic rubbery polymeric materials, which are characterized by stress–strain curves that are almost linear, collagenous tissues have stress-strain curves that exhibit slopes that are also dependent on the orientation of the collagen fibers [41]. For aligned collagenous tissues, such a tendon, the stress-strain curve is almost linear after the low strain planar crimp is removed [41]. The modulus of the collagen molecule is calculated to be about 7 GPa when the slope of the tensile stress-strain curve is corrected for the collagen content and the ratio of the change in the tissue strain divided by the molecular strain [41,42]. For orientable tissues like skin, the value drops to about 4 GPa due to differences in the molecular alignment and the crosslink pattern in skin [42]. Clearly the value of the tissue’s elastic modulus is much lower than that at the molecular level, which is explained by the molecular and fibrillar slippage that occurs during deformation [41,42]. Based on VOCT measurements, it is likely that the collagen fibers in tissue operate at the elbow of the stress-strain curve in vivo [9] under normal loading. When the loads on the tissue are in the upper linear region, plastic deformation and tissue hyperplasia occur similar to that seen when vascular stenosis leads to stiffening of the vessel walls.

### 4.3. Energy Storage and Dissipation by Musculoskeletal and Dermal Tissues

While there have been several recent reports on modeling of the biomechanics of extracellular matrices [43,44], there have been few models describing the molecular mechanism of energy storage. An efficient means to store the energy and transfer it to devices that do work occurs in musculoskeletal tissue. Here the force generating component (muscle) is attached to an energy storage component (collagen fibers) that can then transmit the energy to downstream components (joints) that do musculoskeletal work. The energy storage ability of tendon involves decreasing the entropy that is associated with stretching sections of the collagen molecule that are not in a rigid triple helical conformation [2], as is illustrated in Figure 10. The ends of the triple helix are rich in the amino acid residues proline and hydroxyproline, which form a tight triple helix. These tight triple helices are connected to regions that are more flexible due to the presence of limited amounts of proline and hydroxyproline residues [1]. Upon stretching, uncoiling of these flexible regions occurs, and they adopt a conformation that more closely resembles the rigid parts of the triple helix (see Figure 11). In this manner, elastic energy storage occurs by reversibly decreasing the free energy and entropy of the collagen triple helix upon stretching.

Another example of elastic energy storage occurs in the mammalian cardiovascular system. Elastic energy is stored during pumping and expansion of the arteries that are pressurized by the left ventricle of the heart. The walls of the elastic arteries contain collagen and elastic fibers that are extended to store kinetic energy as potential energy as the heart pumps blood through them [7]. As the pressure falls, when the aortic heart valve closes, the pressure is maintained using work done as the artery wall retracts, pumping blood. During this process the collagen and elastic fibers decrease their vascular circumference. This process involves connections between collagen and elastic fibers that form separate networks in the vessel wall [7].

Similar to the muscle-tendon unit, skin stores applied energy via stretching of the flexible regions of the collagen triple helix. Skin dissipates the stored energy by reversibly “squeezing out” the fluid and proteoglycans found in the interfibrillar matrix [41]. This occurs once the collagen fibers begin to align (see Figure 9) with the loading direction. This viscous dissipation prevents the skin from mechanical failure especially in areas of the body where it covers joints and bones.

## 5. Conclusions

The ability to analyze energy storage and dissipative properties of composite materials helps us not only understand nature’s design for storage devices (tendons) but also the ability of tissues such a skin to dissipate energy through viscous mechanisms.While polymeric materials can reversibly store energy by loss of randomness (decreased entropy), they fail to be able to dissipate energy effectively and ultimately fail due to fatigue and molecular chain breakage.Composite tissues, such as muscle, tendon and skin, store and dissipate energy through entropic changes in collagen (energy storage) and viscous lodissipation) by muscle fibers through heat generation or through rearrangement of the interfibrillar matrix in skin.Energy storage and dissipation by composite materials can be evaluated using vibrational optical coherence tomography (VOCT) non-invasively and non-destructively.

## Figures and Tables

**Figure 1 sensors-21-02001-f001:**
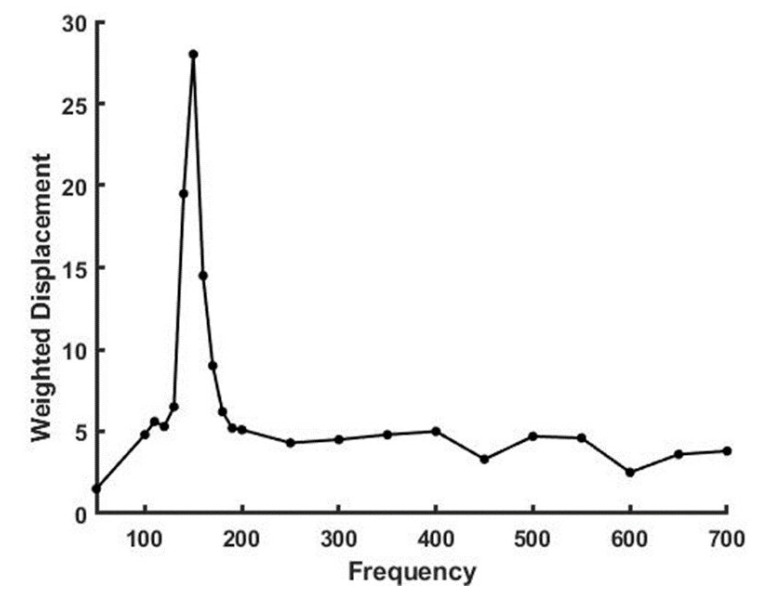
Vibrational analysis of decellularized dermis in vitro composed primarily of type I collagen. The modulus is determined from the frequency at which the maximum displacement is observed and the sample thickness using Equation (1).

**Figure 2 sensors-21-02001-f002:**
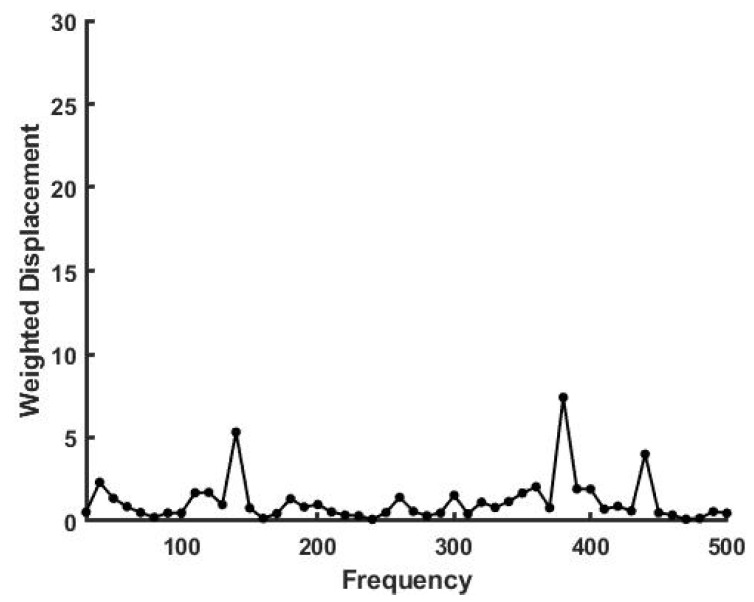
Weighted displacement in μm vs frequency in Hz for skin in vivo over the biceps muscle. Skin is a composite material containing cells, collagen fibers, blood vessels and subcutaneous components including muscle.

**Figure 3 sensors-21-02001-f003:**
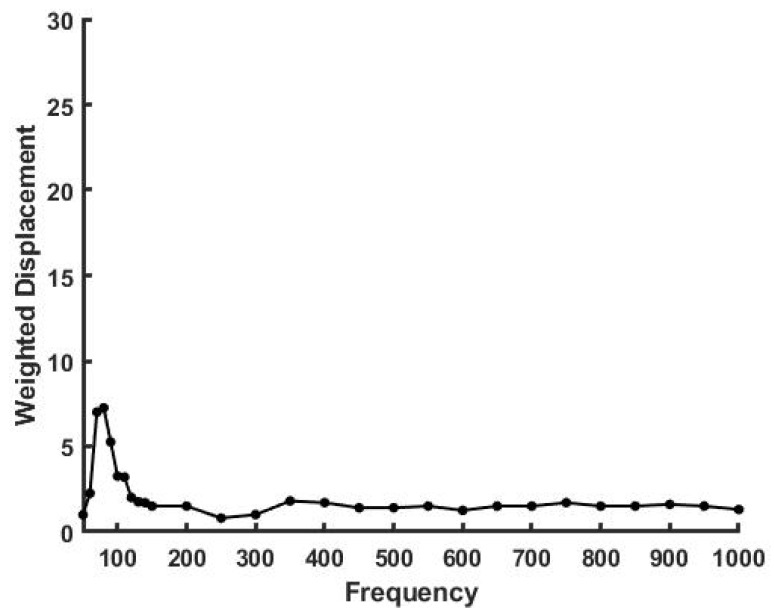
Weighted displacement in μm vs. frequency in Hz for silicone rubber. Note the presence of a single peak at 120 Hz suggesting that the material is monodisperse. The modulus of this sample is 1.7 MPa.

**Figure 4 sensors-21-02001-f004:**
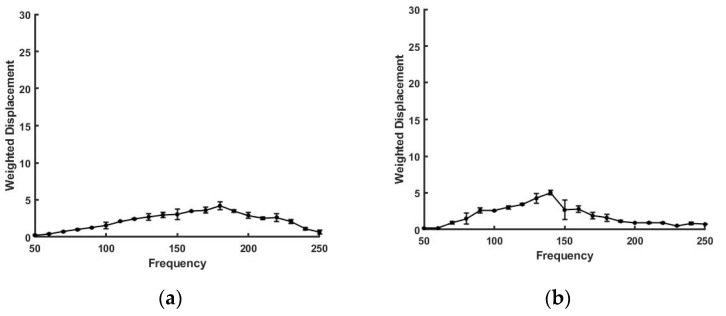
Weighted displacement in μm vs. frequency in Hz for a Viton rubber gasket. Data are shown before (**a**) and after (**b**) high-temperature exposure in an automotive transmission. Note the increased breadth of the resonant frequency peak as a function of frequency reflecting the large molecular weight distribution of the crosslinked polymer chains compared to silicone rubber shown in Figure 3.

**Figure 5 sensors-21-02001-f005:**
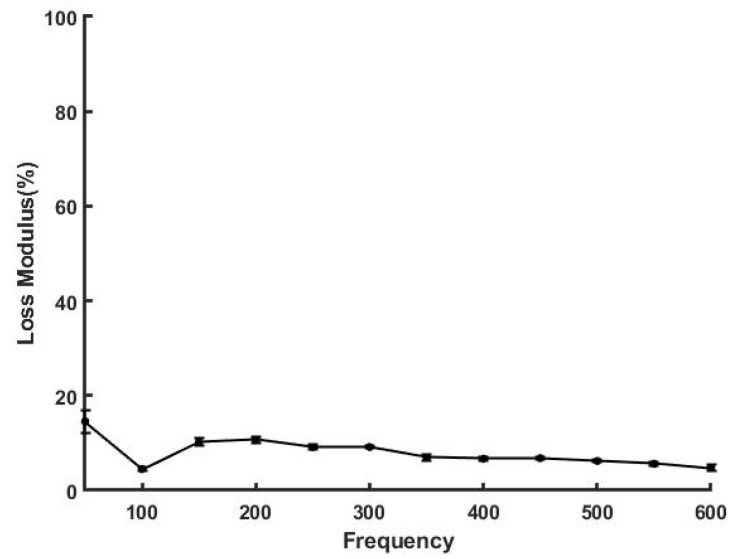
Loss modulus as a % of the elastic modulus for decellularized human dermis in vitro. Note the loss modulus is 12% at low frequencies and drops to about 3% at the resonant frequency.

**Figure 6 sensors-21-02001-f006:**
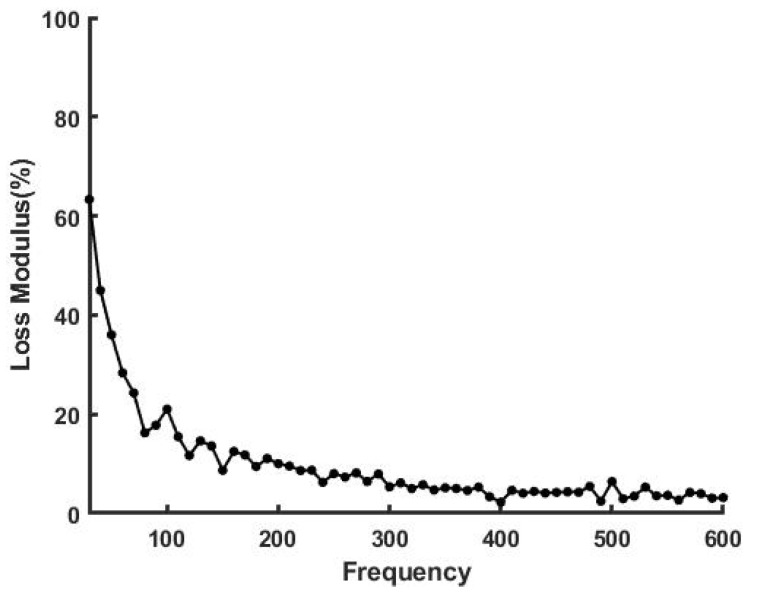
Loss modulus as a % of the elastic modulus for human skin over the biceps muscle in vivo. Note the loss modulus is about 65% at 30 Hz and drops to about 20% at the resonant frequency of collagen. The resonant frequency of cells is between 50 to 70 Hz (Table 1).

**Figure 7 sensors-21-02001-f007:**
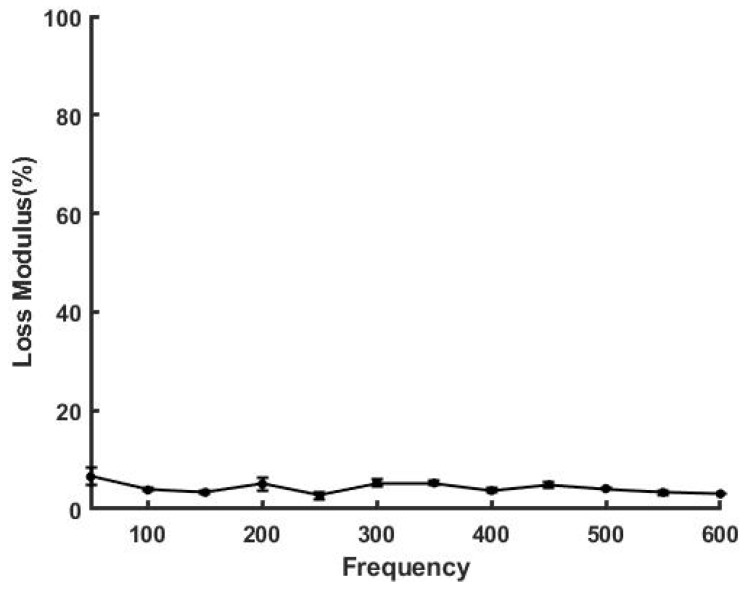
Loss modulus as a % of the elastic modulus for silicone rubber. Note the loss modulus of silicone rubber is only about 4% at most frequencies.

**Figure 8 sensors-21-02001-f008:**
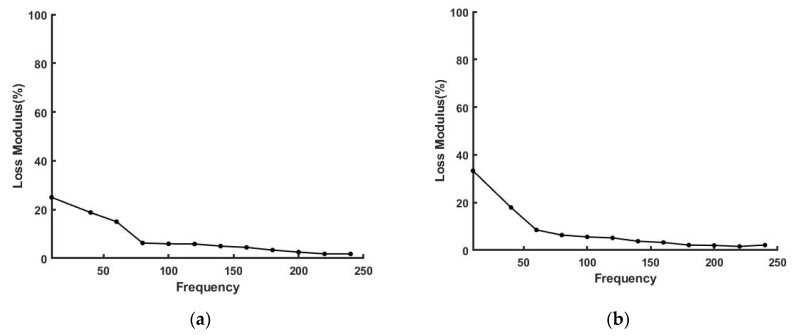
Loss modulus as a % of the elastic modulus for new (**a**) and old (**b**) Viton gaskets. Note the loss modulus increases from about 25% to 35% after cycling of a Viton rubber gasket.

**Figure 9 sensors-21-02001-f009:**
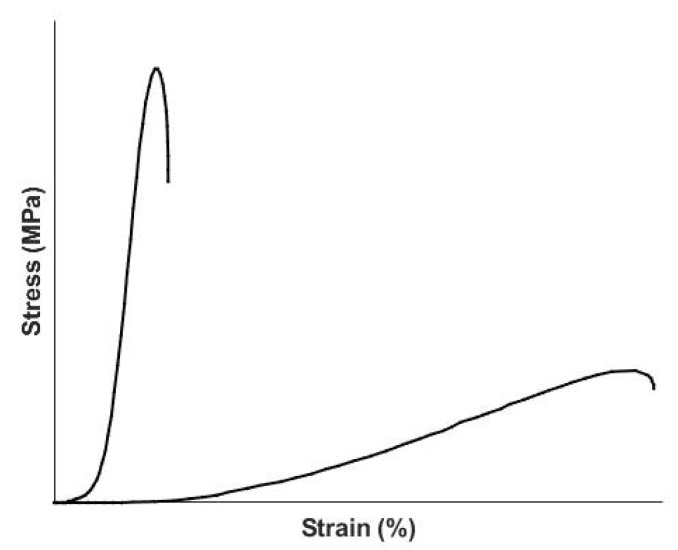
Diagrammatic representation of the stress-strain curves for different extracellular matrices containing collagen fibers. The curve on the left represents the behavior of tendon which contains collagen fibers that run almost parallel to the axis of the tendon. In contrast, skin exhibits a long low modulus region associated with stretching of the elastic fibers and a high modulus region associated with alignment of the collagen fibers with the loading direction.

**Figure 10 sensors-21-02001-f010:**
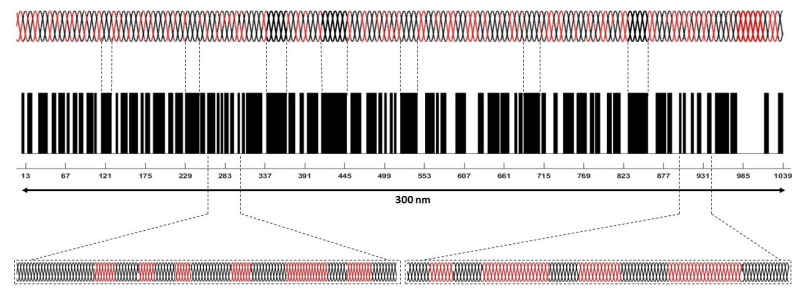
Diagram of the flexible (red) and rigid regions (black) that make up the collagen molecule (top). The collagen triple helix is composed of three left-handed polypeptide chains that are wound around each other to form a right-handed triple helix depicted at the top. The black regions in the bar code below the triple helix (middle diagram) depict the rigid regions containing the amino acid residues, proline and hydroxyproline (black). The white areas represent the flexible regions that are poor in proline and hydroxyproline. Stretching of the triple helix first occurs in the white flexible regions.

**Figure 11 sensors-21-02001-f011:**
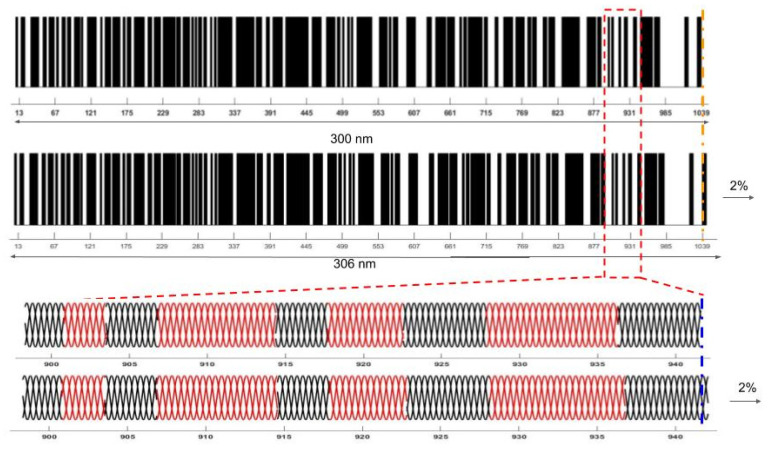
Diagram of energy storage in the collagen molecule via stretching of the flexible regions. When collagen triple helices are stretched, the flexible regions elongate and become more rigid by adopting a conformation more like the rigid regions. The diagram at the top shows the bar code depicting the collagen triple helix with rigid (black) and flexible (white) regions before and after stretching 2%. The diagram below the bar code shows that the red flexible regions deform axially while the black rigid regions are fixed during stretching. The stretching of the flexible regions lowers the entropy of the triple helix. Energy is stored by raising the free energy of the system during stretching while lowering the entropy. That energy is returned reversibly returned to the surrounding tissues when the load is removed from the collagen fibers.

**Table 1 sensors-21-02001-t001:** Resonant frequency and modulus values for natural and synthetic polymers from [9]. Resonant frequencies (Hz), standard deviations (SD) and moduli (MPa) are determined from both in vivo and in vitro testing.

Tissue	Resonant Frequency (Hz) {SD}	Modulus E (MPa) {SD}
Bone		
Lamellar Bone	990 {10.00}	173 {20}
Subchondral Bone	586 {26.07}	67.81 {11.11}
Ear and Lower Nasal cartilage	290 {14.14}	16.2 {1.74}
Upper Nasal Cartilage	380 {14.14}	30.4 {5.89}
Fat, Epidermal Cells	40–70 {12.90}	1.110 {0.25}
Fibrotic Tissue	210 {10}	10.84 {2.48}
Ligament		
Anterior Cruciate Ligament (ACL)	525 {7.07}	53.9 {2.25}
Meniscus	430 {14.14}	31.4 {3.37}
Muscles		
Bicep Muscle	378 {16.02}	29.6 {2.62}
Quadriceps Muscle	365 {21.21}	20.5 {2.32}
Nerve		
Nerve	266 {11.54}	15.86 {2.24}
Normal Skin	110 {7.38}	2.15 {0.29}
Ocular		
Cornea, Sclera	140 {14.14}	2.4 {0.14}
Tendon		
Achilles Tendon	440 {10.00}	34.0 {5.98}
Flexor Digitorum Profundus Tendon	370 {14.14}	22.7 {9.42}
Patellar Tendon	430 {5.77}	33.8 {4.62}
Vascular		
Carotid Artery	136 {11.54}	4.64 {0.98}
Radial Artery	155 {11.98}	3.66 {0.65}
Vein	165 {7.07}	4.84 {0.025}
Sample	Resonant Frequency (Hz) {SD}	Modulus E (MPa) {SD}
ABS Plastic	2800 {10}	2120 {0.02}
Silicone Rubber	80 {10.00}	1.68 {0.23}
New Viton Gasket	180 {5}	11.45 {0.64}
Old Viton Gasket	140 {5}	5.99 {0.43}

## Data Availability

Data is available to share with interested parties on request.

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
