# Peer review of "Use of Vibrational Optical Coherence Tomography to Analyze the Mechanical Properties of Composite Materials"

_sensors, 2021, doi:10.3390/s21062001_

Round 1
Reviewer 1 Report
This paper reviewed the elastic energy storage and dissipation for biological tissues and synthetic materials evaluated by vibrational optical coherence tomography. VOCT can noninvasively measure the displacement of frequency that defines the characteristics of structural component of tissue or materials. I think this paper is suitable for publication in Sensors.
Reviewer 2 Report
Paper could be improved a lot. Check suggestions in attached fle

Reviewer 3 Report
The author utilized OCT to observe the vibration of particles produced by audible sound as the excitation source and was looking for the resonant frequency in various tissues once the particle displacement reaches to the maximum. The study is interesting. The reviewer just has a few questions showed below.
- Compared with the previous paper by the author, the author just enlarged the database in this manuscript. The reviewer has a little bit curious about the novelty in this manuscript compared with previous published papers by the author.
- The study method has been provided in previous paper; however, the author still needs to mention certain important information in this manuscript, for example the system structure, OCT scan rate, what kind of scan methods and so on.
- The wavelength of OCT is small (I guess it could be 1300nm?), did the author observe any phase-wrapping situations? If so, the author should describe how to solve this issue in the manuscript. If no, how does the author know that the particle displacements have been reached to the maximum?
Round 2
Reviewer 2 Report
I see a clear improvemnt of the work, now it could be interesting for reading
Reviewer 3 Report
I am satisfied with the authors' responses.